# Early Transcriptional Changes of Adipose-Derived Stem Cells (ADSCs) in Cell Culture

**DOI:** 10.3390/medicina58091249

**Published:** 2022-09-09

**Authors:** Sara Taha, Elif Akova, Maximilian Michael Saller, Riccardo Enzo Giunta, Elisabeth Maria Haas-Lützenberger

**Affiliations:** 1Department of Orthopaedic, Trauma and Rehabilitative Medicine, Musculoskeletal University Center Munich, Ludwig-Maximilians-University (LMU), Fraunhoferstraße 20, 82152 Planegg-Martinsried, Germany; 2Division of Hand, Plastic and Aesthetic Surgery, Ludwig-Maximilians-University (LMU), Ziemssenstr. 5, 80336 Munich, Germany

**Keywords:** adipose-derived stem cells, cell isolation, cell culture, cell sequencing

## Abstract

*Background and Objectives*: While autologous fat grafting has been carried out in the clinical field for many years, the utilization of isolated and cultured adipose-derived stem cells (ADSCs) is highly restricted in many countries. However, ADSCs are under investigation currently and heavily researched in many cell-based therapy approaches in the field of regenerative medicine. Objective: For the utilization of future cell-based therapies with ADSCs, in vitro cell expansion might be necessary in many cases. Thus, the cellular characteristics of ADSCs may be altered though the process of being cultured. The aim of this study was to assess changes in the gene expression profile of ADSCs after cell expansion for 48 h. *Materials and Methods*: Isolated ADSCs from five different donors were used for in vitro expansion. For the evaluation of the gene expression profile, mRNA deep Next-Generation Sequencing was performed to evaluate the differences between cultured and freshly isolated cells. *Results*: Our study gives insight into transcriptional changes in ADSCs after a short cell cultivation period. This includes the most prominent upregulated genes such as PPL, PRR15, CCL11 and ABCA9, as well the most downregulated genes, which are FOSB, FOS, EGR1 and DUSP6. Furthermore, we showed different biological processes that changed during short-term cell expansion, which led to downregulation of fat-associated metabolism hormone processes and to an upregulation of extracellular matrix-associated genes. *Conclusion*: In conclusion, our study reveals a detailed insight into early changes in the gene expression profile of cultured ADSCs. Our results can be utilized in future experiments.

## 1. Introduction

During the last two decades, a growing interest has been expressed in the use of adult multipotent somatic stem cells, which reside in white fat tissue (WAT) and are also defined as adipose-derived stromal cells (ADSCs). These cells can be differentiated into several cell lineages, including adipocytes, chondrocytes and osteocytes [1,2]. In contrast to alternative tissue sources of adult stem cells such as bone marrow-derived cells, ADSCs can be harvested in large numbers by minimally invasive procedures (i.e., liposuction), and thus have a low donor-site morbidity [3,4]. Consequently, these cells are of great interest in regenerative medicine and surgery. Numerous studies have been conducted regarding the regenerative potential of these cells [5,6,7,8,9]. Since many publications about the positive characteristics of these cells have been published, their utilization in regenerative medicine has grown tremendously in a broad variety of different medical fields [10]. Because of their mesodermal origin, the clinical application of these cells seems multitudinous. Areas of clinical interest range from musculoskeletal regeneration, cardiovascular tissue regeneration and nerve repair to immunomodulatory effects of ADSCs [11,12,13,14,15]. Extraction of ADSCs involves the isolation of the stroma-vascular fraction (SVF) from lipoaspirate and/or solid WAT. ADSCs are then isolated from the SVF [2,5,9,16]. Despite the growing number of clinical trials, ADSCs are restricted for clinical application in many countries. To date, there are no clinically approved ADSCs-based products. Challenges for clinical translation of cell therapies mainly lie in the cultivation and basic characterization of ADSCs.

The process of isolation and the subsequent culture procedure of ADSCs are not standardized, portraying discrepancies with previously published results. While several authors assess the possible effects of the isolation technique on viability and composition of ADSCs, only a limited number of studies explore the derived outcome of the cultivation procedure on ADSC characteristics [10,16,17,18,19]. In addition to the material used in culture dishes, as well as culture medium, data indicate that the cultivation period in particular affects the character of ADSCs. Studies demonstrate a significant variation in surface marker expression and thus the phenotype of ADSCs with successive passaging [16,20,21,22]. Interestingly, possible alterations in gene expression during cell culture of ADSCs remain to be determined. Therefore, this study set out to assess possible changes present. For this purpose, we used next-generation mRNA sequencing to analyze gene expression changes of ADSCs directly after isolation and after a 48 h incubation period.

## 2. Materials and Methods

### 2.1. Ethics Statement and Sample Acquisition

Human lipoaspirates were obtained from five patients in a general good health, who received water-jet-assisted liposuction with the Body-Jet system (Human Med AG, Schwerin, Germany) for aesthetic reasons (Table 1). Liposuction with the Body-Jet system was performed either with 3.5 and/or 3.8 mm cannulas and with a pressure of approximately 550 bar. A Tumescent solution containing saline, lidocaine, and epinephrine was infiltrated prior to extraction with the Body-Jet system. The obtained 5 samples of adipose tissue (50 mL) were centrifuged for 5 min and the mid-layer, consisting of tumescent fluid, was extracted before further use of the samples. All patients tested negative for HIV (human immunodeficiency virus), HCV (hepatitis C virus), and HBV (hepatitis B virus). The study was approved by the ethics committee of Ludwig-Maximilian-University, Munich (275-16) and samples were obtained after written informed consent signed by patients.

### 2.2. Cell Isolation and Culture Conditions

For the isolation of ADSCs, approximately 10 g of lipoaspirate was used for enzymatic isolation with a semi-automated centrifuge system (ARCTM-Processing Unit, InGeneron, Houston, TX, USA) followed by the manufacturer’s protocol using the recommended enzyme blend (Matrase™) and lactated Ringer’s solution (Fresenius Kabi, Bad Homburg, Germany) at 37 °C. The time between tissue harvesting and cell isolation was less than 5 h, while the isolation process with the InGeneron centrifuge system took approximately 90 ± 20 min. The obtained SVF (approximately 1.2 × 10^6^ cells) was resuspended and cultured in standard culture medium consisting of DMEM-high glucose (Thermo Fisher Scientific, USA) supplemented with 10% fetal bovine serum (FBS, Sigma-Aldrich, Burlington, MA, USA), 100 U/mL Penicillin and 100 μg/mL Streptomycin (Life Technology, Carlsbad, CA, USA). After 24 h, culture flasks were washed several times with phosphate-buffered saline (PBS) to discard the non-adherent cells. Adherent cells qualified as ADSCs. Subsequently, cells were divided into the two experimental groups: RNA from ADSCs was either directly isolated or after 48 h of culturing in a humidified incubator (21% O_2_, 5% CO_2_ and 37 °C).

In our previous study from 2019, we showed the functional analysis of ADSCs including the 5 donors from this study. The proliferation potential of ADSCs was subsequently assessed by cumulative population doublings (cumPD), population doubling time (PDT), colony-forming units (CFU), and cell metabolism assays. To prove the multipotency of the primary isolated cells, ADSPCs were then induced to differentiate into adipogenic, osteogenic, and chondrogenic lineages [11].

### 2.3. mRNA-Sequencing and Bioinformatics

RNA was isolated with the RNeasy kit (Qiagen, Hilden, Germany) following their standardized protocol. RNA quality was measured with BioAnalyzer (Agilent, Santa Clara, CA, USA) and libraries for sequencing were prepared with a SENSE mRNA-Seq Library Prep Kit V2 (Lexogen, Vienna, Austria). All libraries were sequenced on a HiSeq1500 device (Illumina, San Diego, CA, USA) with a read length of 50 bp and a sequencing depth of approximately 20 million reads per sample. The raw fastq files were then demultiplexed with Illumina_Demultiplex. Transcriptomes were aligned to the human reference genome GRCh38.99 by using STAR (version 2.7.2b) to obtain the reads per gene [23]. All sequencing analysis was performed on R programming (version 4.1.0). Initially, the data were filtered among low row sum of read per gene counts by cut-off value 5. Following this, the data were then filtered using only “protein coding” genes for the next analysis. With the program R results, tables were generated in order to identify statistically significant, differentially expressed genes (DEGs) between the samples before (t0) and after incubation (t1 = 48 h) [24]. The RNA-seq data analysis is calculated by the DESeq2 package (version 1.33.4). To define Gene Ontology Biological Pathways, we used the clusterProfiler (v4.1.3). Using the Benjamini–Hochberg correction for multiple hypothesis testing genes with an adjusted *p*-value < 0.05 was considered as significant. A cut-off of log2FoldChange (log2FC) > 2 and log2FoldChange (log2FC) < −2 was applied to extract differentially expressed genes.

## 3. Results

### 3.1. Identification of Differentially Expressed Genes (DEGs) in ADSCs after Cell Cultivation for 48 h

In order to identify potential transcriptome changes of ADSCs, all five samples were analyzed before and after 48 h of incubation in standard cell culture conditions by mRNA-sequencing (mRNAseq). From 15,107 genes, 101 genes were significantly differentially expressed after treatment. The top 10 up- and downregulated genes were normalized by gene expression and compared among each donor, which is portrayed in MA-Plot format (Figure 1). This represents variation in each gene expression between donors after no incubation in contrast to incubation for 48 h.

The analysis of the most frequently upregulated genes included PPL, PRR15, CCL11 and ABCA9, whilst the most downregulated genes included FOSB, FOS, EGR1 and DUSP6.

The hierarchical analysis of the principal component 1 (PC1) revealed a clear separation of ADSCs cultured for 48 h and uncultured ADSCs. Principal Component Analysis (PCA) shows the clustering between donors depending on the treatment condition. In addition, after cultivation of 48 h, ADSCs showed a separation along PC2 with an increasingly homogeneous response when compared to uncultured ADSCs (Figure 2).

The variance stabilizing transform (VST) normalized logarithmic gene expression used for the PCA calculations (Figure 2). This figure demonstrates that cultured ADSCs have a slightly higher homogenous gene expression profile when compared to uncultured cells.

The top 10 genes that mostly contribute to the PC1 prove the hierarchical clustering due to the treatment excluding for D3 before (Figure 2). This donor (D3) shows an inclined differentiation between two time points when compared to the other four donors (Figure 1). The 10 most accountable genes in PC1, such as FOS, FOSB and EGR1, are important in various cellular processes such as cell proliferation, differentiation and apoptosis. In Figure 3, we show the hierarchical cluster analysis of the 101 significantly differential expressed genes. This represents the normalized gene expression of each donor. It is noticeable that one donor (D3) shows stronger difference between the two time points in contrast to the four remaining donors.

### 3.2. Biological Pathway Analysis

To analyze the behavior of cellular components and biological pathways, we carried out a pathway analysis using Gene Ontology (GO) and included all DEGs that were significantly changed after a 48 h incubation period. The significant 101 genes were then used to define Gene Ontology Biological Pathways and the top five activated and suppressed pathways with the effect of treatment ordered by gene rankings and normalized enrichment score (NES) (Figure 4). The analysis with GO of significantly changed genes in ADSCs after 48 h of cell cultivation, when compared to uncultured ADSCs, showed alterations in different biological pathways, such as regulation of cell morphogenesis, cell–substrate adhesions, glycoprotein metabolic process, and regulation of supramolecular fiber organization. The most significantly downregulated biological functions were the following: response to steroid hormones, response to corticosteroid and glucocorticoid, response to cAMP, and response to hormones (Figure 4).

## 4. Discussion

Since their discovery in the early 2000s, ADSCs have been seen as a promising therapy tool in the field of tissue engineering and regenerative medicine. Despite the great clinical interest in ADSCs for different therapy approaches, the use of isolated stem cells is not clinically approved in most countries. Therefore, the procedure of transferring fat (autologous fat grafting) is a surgical technique used worldwide for many years. There is a multitude of scientific questions regarding the long-term effects of fat transfer and ADSCs that, as of yet, have not been fully answered. Therefore, in vitro studies will continue to play a significant role in ADSC studies. The aim is to provide molecular knowledge of ex vivo properties of ADSCs and portray basic knowledge in purification and application of those cells in a clinical setting. Various studies have shown that the characteristics of ADSCs can become altered during prolonged culturing [25,26]. However, there is not much yet known about transcriptional gene expression changes during short-term culturing. In our study, we identified gene expression regulations in ADSCs before and after cell cultivation for 48 h, as well as pathways induced or repressed after cultivation that may indicate in which way cells may modulate the tissue according to the location to which they were transplanted.

After correction for multiple testing, 101 genes showed significant differential gene expression in ADSCs after cultivation when compared to ADSCs before cell culturing. Out of these significantly changed genes, 12 genes were up- and 89 were downregulated.

In our analysis, the top-regulated biological pathways in ADSCs after cell cultivation of 48 h are regulation of cell morphogenesis, cell–substrate adhesions, glycoprotein metabolic process, and regulation of supramolecular fiber organization (Figure 4). Furthermore, we displayed the most significantly downregulated biological functions, which are response to steroid hormones, response to corticosteroid and glucocorticoid, response to cAMP, and response to hormones. Changes in gene expression can prevent cells from responding to mitogenic signals, affect reproduction, and even alter their metabolic status [27]. To summarize, there is no conspicuous pathway after incubation of 48 h affected. The upregulated pathways were those one would expect after 48 h of cultivation. The changes in cell cycle differentiation, metabolic processes, or extracellular matrix behavior are no pathways that lead to a certain function. One can summarize that fat cells lose their “fingerprint” over time and further develop cytoskeletal dynamics and extracellular matrix deposition. This is an expected behavior of these cells after cultivation.

In our study, we identified the top up and downregulated genes in ADSCs after a cultivation period of 48 h. *PPL* is the top upregulated gene that belongs to the plakin families, which are known for molecular bridges in terms of linking intracellular cell–cell junctions and the cytoskeleton. This protein is generally localized in desmosomes and interdesmosomal plasma membranes of differentiated epidermal keratinocytes. It also acts as an adhesion molecule, which plays a role in cellular movement [28,29].

The second most upregulated gene is *ABCA9*. It is a member of the ATP-binding cassette (ABC) transporter superfamily, which includes seven subfamilies that participate in the ATP transport of various substances across membranes. Besides the role of ABC transporters in chemoresistance, through the drug efflux from cancer cells, recently published studies have found an additional role also in tumor initiation and tumor progression [30,31]. Whereas some ABC transporters are more discussed in the literature, ABCA9 is not. Interestingly, a study could show that ABCA9 is involved in mononuclear cell differentiation and lipid transportation [32,33,34].

The third top upregulated gene is Prolinerich15 (*PRR15*), which belongs to a small family of genes encoding prolinerich proteins (PRR) [35,36]. Its role in proliferation and differentiation is also discussed [1]. Furthermore, in a recent paper, Wang et al. showed that it is upregulated in esophageal cancer cells [2].

Lastly, the sixth most upregulated gene is known as C-C motif chemokine 1/eosinophil-selective chemokine/ eotaxin-1 “*CCL11*”. It belongs to the chemokine family “*CC*” and, as the name implies, is potent for eosinophils. It recruits eosinophils by causing an induction of chemotaxis, which is a prominent feature in allergic reactions. A recent paper from Kindstedt et al. showed that both in vitro and in vivo osteoblasts express *CCL1*1 when an inflammatory bone lesion is present. Chemokines are known for their ability to regulate bone metabolism [37,38]. Kindstedt et al. showed that *CCL11* stimulates bone resorption, and thus hypothesized that it could be of importance for migration of pre-osteoclasts to bone surfaces.

The first and the second significantly downregulated genes belong to the same protein family “FOS”. The transcription factor FOS is known as a proto-oncogene and therefore a regulator of cell proliferation, differentiation and transformation [39,40]. FOSB also belongs to the FOS family but has been considered as a mechano-responsive gene that stimulates an osteogenic differentiation program in osteoblasts in response to mechanical stress and, moreover, plays a role in regulation of osteoblast proliferation and differentiation [41]. The third significantly downregulated gene is EGR, which belongs to the early growth response protein (EGR) family and functions as an oncogene and thus as a transcriptional regulator. EGR is important in various cellular processes, such as cell proliferation, differentiation, and apoptosis. Epidermal growth factor receptor (EGFR) activates ERG1 and thus is an important regulator in controlling key cell cycle regulators, cytokines, and co-stimulating molecules [41].

Genes such as Matrix metalloproteinase-3 (*MMP3*) are enzymes that are involved in the breakdown of extracellular matrix proteins, for example in disease processes such as osteoarthritis [42]. MMP3 is increased in different processes of the body wherever inflammation is present [43]. We showed that after an incubation period of 48 h, MMP3 was significantly downregulated.

To our surprise, the top up and downregulated genes are not associated with processes that are known to be important in the field of lipofilling and lipotransfer. Our aim was to show in an unbiased manner the top regulated genes. Therefore, we are discussing only the relevant genes, even though they seem not to have an impact on lipofilling and cell-assisted lipotransfer.

In conclusion, the present study gives a broad insight of changes in the gene expression profile of cultured ADSCs and can be utilized in future experiments, by displaying genes, which might be fundamental for different cell-based therapies. A limitation of the study is the fact that only one time point (48 h) of cell culturing was analyzed. Thus, our results portrayed no significant difference between long- or short-term incubation time amongst adipose cells.

Furthermore, another limitation of our study is the small number of five patients included in the study. For a deeper understanding, the study should be repeated with a higher patient number.

## Figures and Tables

**Figure 1 medicina-58-01249-f001:**
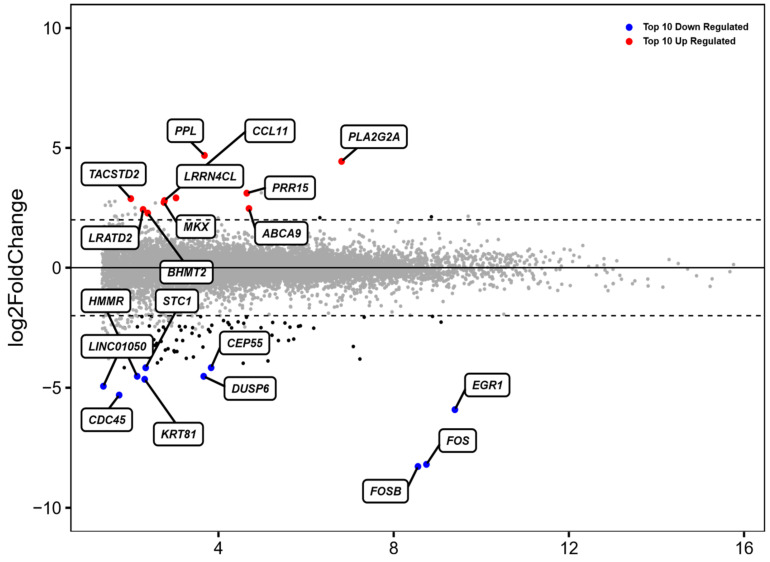
MA plot of the top 10 up- and downregulated (out of 101) differentially expressed genes (DEGs). The blue points represent downregulated and red points represents upregulated genes. The horizontal dashed line represents the statistical thresholds of DEGs (adjusted *p*-value < 0.05 and |fold change| > 2).

**Figure 2 medicina-58-01249-f002:**
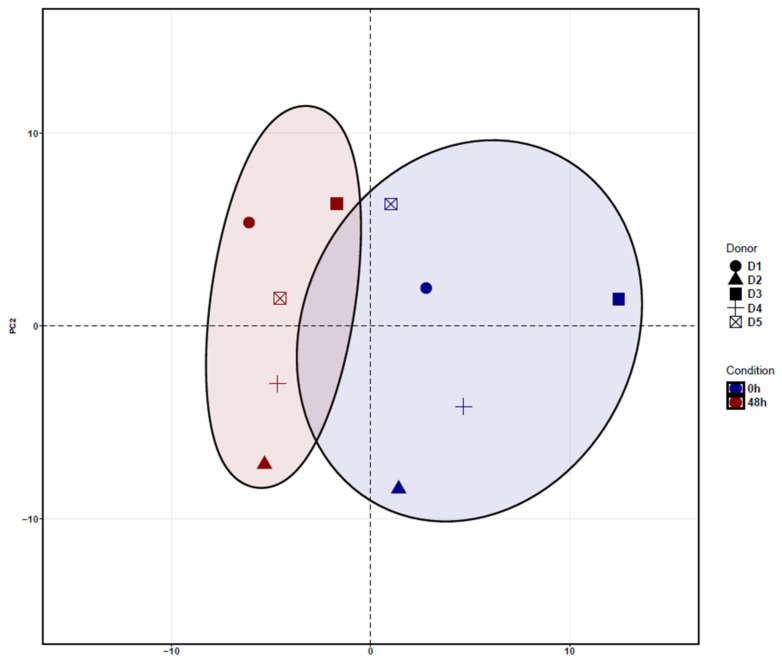
Principal Component Analysis (PCA) of cultured ADSCs for 48 h and uncultured ADSCs. The PCA revealed a clear clustering of cultured ADSCs (blue dots) and uncultured ADSCs (red dots) along the main component PC1. In addition, after cultivation of 48 h, ADSCs showed a separation along PC2, with a more homogeneous response, when compared to uncultured ADSCs.

**Figure 3 medicina-58-01249-f003:**
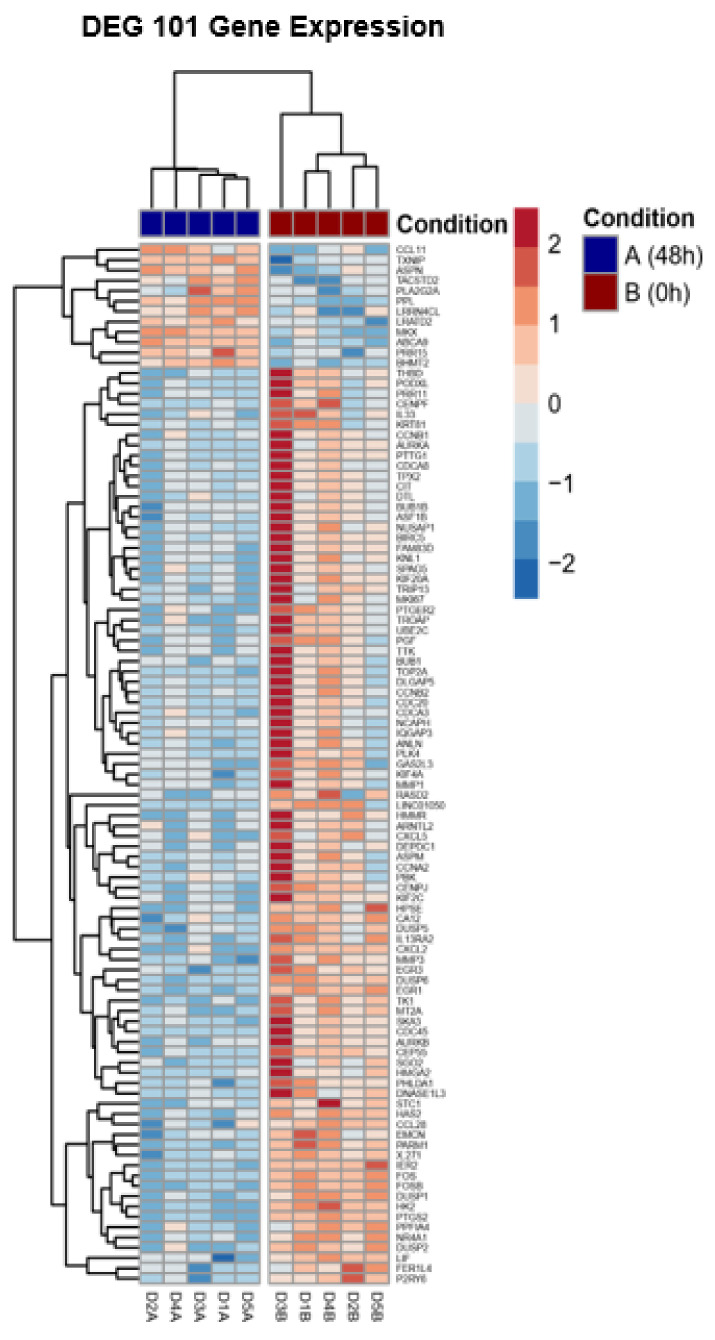
Hierarchical cluster analysis of the 101 significant differentially expressed genes. It represents normalized gene expression of each donor.

**Figure 4 medicina-58-01249-f004:**
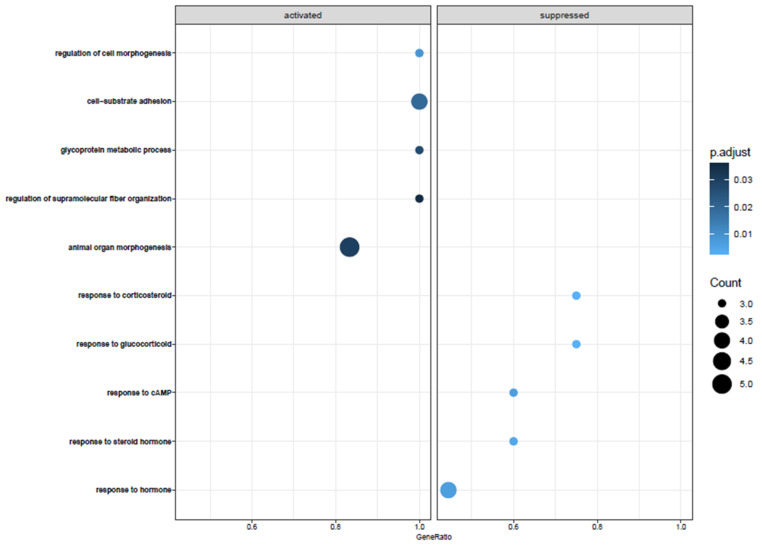
The GeneOntology analysis of all significant 101 genes that are differentially expressed between ADSCs after 48 h of incubation and uncultured ADSCs. Activated and suppressed pathways Normalized Enrichment Score (NES) is calculated by the LogFoldChange values. It revealed change in a plentitude of different biological functions, such as downregulation of fat-associated metabolism hormone processes and an upregulation of extracellular-matrix-associated genes.

**Table 1 medicina-58-01249-t001:** Patient information.

Patient Overview	Sex	Age	Harvesting Site	BMI	Surgery Type
donor 1	female	56	abdomen	24.7	aesthetic
donor 2	female	76	thighs	35.5	aesthetic
donor 3	male	25	abdomen	23.2	aesthetic
donor 4	female	26	thighs	21	aesthetic
donor 5	male	38	abdomen	27.8	aesthetic

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
