# Peer review of "Early Transcriptional Changes of Adipose-Derived Stem Cells (ADSCs) in Cell Culture"

_medicina, 2022, doi:10.3390/medicina58091249_

Round 1

Reviewer 1 Report

In the manuscript entitled „Early transcriptional changes of adipose-derived stem cells (ADSCs) in cell culture“, the authors present differences in the expression profile induced by standard cell culture condition in comparison to cells cultured only until adherence. Because fat grafting is a method increasingly used in regenerative medicine, because there is an ongoing debate about regulatory restrictions for the use of autologous stem cells, and because little is known about transcriptional changes of ADSCs during cell culture the study is important and topical. However, some concerns have to be addressed:

General remarks:

The study is very descriptive (as in “not hypothesis-driven”). Whereas it should be possible to publish descriptive results, the discussion needs to make up for this “flaw”. The manuscript would benefit from discussing possible implications of the differentially expressed genes and the pathways for autologous fat transfer. The discussion lacks a focus on the implications of cell culturing for their application in regenerative medicine. It seems that the discussion of the genes and pathways found to be regulated in the cultured cells has absolutely nothing to do with lipofilling, cell assisted lipotransfer or regenerative medicine whatsoever. Without the first paragraph, no one reading the discussion would come to the conclusion that autologous fat grafting is relevant to the paper.

5 patients is a relatively small number of patients. Whereas it should be possible to publish these results, the small number should be mentioned as a limitation of the study.

All figures show a low resolution. Most gene names are unreadable. Please provide higher resolution images.

The whole manuscript needs a language check. I know, I know, there is always the “revise language” comment from every reviewer as soon as the authors aren’t native speakers. However, there are several dubious punctuations and some really confusing phrasings. For example, in lines 344ff, it says “Genes as Matrix metalloproteinase-3 (MMP3), which are enzymes that are involved in breakdown of extracellular matrix proteins, for example in disease processes like osteoarthritis [43].” That is not a complete sentence. I know that the “revise language” comments are annoying, but You REALLY need to revise the language of the manuscript. I am sorry.

Minor issues:

In the introduction (line 70), the authors claimed “to analyze gene expression changes of ADSCs directly after isolation […]”. However, according to the material and method section, the control ADSCs were cultured overnight until adherence prior to RNA isolation. Therefore, the authors should rephrase “directly after isolation” to something less contradictory.

Line 78: “[…] with a pressure of approximately 550 bar.” 550 bar is the pressure at a depth of 5,500 meters underwater, halfway down to the mariana trench…

Line 80: “[…] and the mid-layer, consisting of tumescent fluid […].” In my experience, the undermost layer contains the tumescent fluid. Please rephrase or explain what layer is beneath the aqueous layer.

Line 283: “[…] shot-term culturing.” An “r” is missing here…

The discussion section in lines 296-304 is somewhat confusing. What is a specific pathway? Why are the pathways mentioned not specific? What is the purpose of the first sentence I this section? Why can one summarize that the cells are losing their fingerprint? Please elaborate.

Line 307: “[…]  in terms of liking intracellular cell-cell junctions […].” Is there a “n” missing?

Line 321 (and 327): “[…] in a recent paper Wang et al showed […].” There should be a dot after “al”.

Author Response

Dear ladies and gentlemen,

We would like to thank the Editorial Office and the reviewers for their valuable comments concerning our manuscript entitled “Early transcriptional changes of adipose-derived stem cells (ADSCs) in cell culture”. We have carefully read your and the reviewer’s suggestions, which were very helpful in further improving and focusing our manuscript. According to those valuable remarks, we have adapted our manuscript and would like to resubmit it for reconsideration. In the following, we list the reviewer’s comments point-by-point along with our responses (blue) and the corresponding changes we have made to the manuscript.

Points raised by reviewer 1:

  1. The study is very descriptive (as in “not hypothesis-driven”). Whereas it should be possible to publish descriptive results, the discussion needs to make up for this “flaw”. The manuscript would benefit from discussing possible implications of the differentially expressed genes and the pathways for autologous fat transfer. The discussion lacks a focus on the implications of cell culturing for their application in regenerative medicine. It seems that the discussion of the genes and pathways found to be regulated in the cultured cells has absolutely nothing to do with lipofilling, cell assisted lipotransfer or regenerative medicine whatsoever. Without the first paragraph, no one reading the discussion would come to the conclusion that autologous fat grafting is relevant to the paper.

We thank the reviewer for the supportive comment and accordingly added an explanation to the Discussion section.

  1. 5 patients is a relatively small number of patients. Whereas it should be possible to publish these results, the small number should be mentioned as a limitation of the study.

We thank the reviewer for this comment. The aim of our work is big data, a patient number of 20 to 30 patients would for sure give us more information. However, this study is the first step for our work in this field, therefore we were not able to perform NGS with more patients sample at that time. At this point group clustering is the main focus in our study to keep it clinical relevant. In the future we aim to work with more biological replicated, to find more patient specific characterization.

  1. All figures show a low resolution. Most gene names are unreadable. Please provide higher resolution images.

We thank the reviewer for this comment. We exchanged the figures in higher resolution.

  1. The whole manuscript needs a language check. I know, I know, there is always the “revise language” comment from every reviewer as soon as the authors aren’t native speakers. However, there are several dubious punctuations and some really confusing phrasings. For example, in lines 344ff, it says “Genes as Matrix metalloproteinase-3 (MMP3), which are enzymes that are involved in breakdown of extracellular matrix proteins, for example in disease processes like osteoarthritis [43].” That is not a complete sentence. I know that the “revise language” comments are annoying, but You REALLY need to revise the language of the manuscript. I am sorry.

We thank the reviewer to raise this crucial point. We checked the whole manuscript for spelling mistakes and improved the language.

  1. In the introduction (line 70), the authors claimed “to analyze gene expression changes of ADSCs directly after isolation […]”. However, according to the material and method section, the control ADSCs were cultured overnight until adherence prior to RNA isolation. Therefore, the authors should rephrase “directly after isolation” to something less contradictory.

Line 78: “[…] with a pressure of approximately 550 bar.” 550 bar is the pressure at a depth of 5,500 meters underwater, halfway down to the mariana trench…

Line 80: “[…] and the mid-layer, consisting of tumescent fluid […].” In my experience, the undermost layer contains the tumescent fluid. Please rephrase or explain what layer is beneath the aqueous layer.

Line 283: “[…] shot-term culturing.” An “r” is missing here…

The discussion section in lines 296-304 is somewhat confusing. What is a specific pathway? Why are the pathways mentioned not specific? What is the purpose of the first sentence I this section? Why can one summarize that the cells are losing their fingerprint? Please elaborate.

Line 307: “[…]  in terms of liking intracellular cell-cell junctions […].” Is there a “n” missing?

Line 321 (and 327): “[…] in a recent paper Wang et al showed […].” There should be a dot after “al”.

We thank the reviewer for this helpful comments. We checked the whole manuscript for spelling mistakes and improved the language.

Reviewer 2 Report

The authors cultured SVF from five patients who had undergone liposuction as a cosmetic measure for 48 hours in culture medium, and analyzed and evaluated the genes that changed before and after the culture of ADSCs by mRNA deep Next Generation Sequencing.

 The purpose of analyzing the early changes in gene expression profiles of cultured ADSCs is to provide molecular knowledge of the in vitro properties of ADSCs and to provide fundamental knowledge for the purification and clinical application of these cells. The characteristics of ADSCs have been shown to change during long-term culture by various studies, but the changes in transcriptional gene expression during short-term culture are still not well known.

The authors found that PPL, PRR15, CCL11, and ABCA9 were upregulated in ADSCs before and after 48 hours of cell culture, while FOSB, FOS, EGR1, DUSP6 genes were downregulated. The authors showed different biological processes that changed during short-term cell expansion. The method of analysis and evaluation of the results is appropriate.

I understand that mRNA deep Next-Generation-Sequencing is expensive and that the number of ADSCs is limited, however, if there were your results not only of gene expression but also of cell morphology changes due to differentiation, this would be more appreciated.

As the authors conclude, five different ADSCs showed similar gene expression trends in a "short incubation period" of 48 hours. This result seems to suggest that this ADSCs culture method could be one of the criteria.

A  minor revision is listed below.

1.     Line 283 ( 9 of 13), “shot-term “may be “short”.

Author Response

(The authors gave the same response as above.)
